# Significance of the Gut Microbiota in Acute Kidney Injury

**DOI:** 10.3390/toxins13060369

**Published:** 2021-05-22

**Authors:** Taku Kobayashi, Yasunori Iwata, Yusuke Nakade, Takashi Wada

**Affiliations:** 1Department of Nephrology and Laboratory Medicine, Kanazawa University, Kanazawa 920-1192, Japan; takuwgmania@hotmail.com (T.K.); nakadeyukeitu@gmail.com (Y.N.); twada@m-kanazawa.jp (T.W.); 2Division of Infection Control, Kanazawa University Hospital, Kanazawa 920-1192, Japan

**Keywords:** gut microbiota, acute kidney injury, D-amino acid

## Abstract

Recent studies have revealed that the gut microbiota plays a crucial role in maintaining a healthy, as well as diseased condition. Various organs and systems, including the kidney, are affected by the gut microbiota. While the impacts of the gut microbiota have been reported mainly on chronic kidney disease, acute kidney injury (AKI) is also affected by the intestinal environment. In this review, we discussed the pathogenesis of AKI, highlighting the relation to the gut microbiota. Since there is no established treatment for AKI, new treatments for AKI are highly desired. Some kinds of gut bacteria and their metabolites reportedly have protective effects against AKI. Current studies provide new insights into the role of the gut microbiota in the pathogenesis of AKI.

## 1. Overview of the Gut Microbiota

Over 1000 genera of bacteria colonize the human intestine and shape the composition of the gut microbiota [1]. Most of the bacteria in the human intestine belong to four phyla, *Firmicutes*, *Bacteroidetes*, *Proteobacteria*, and *Actinobacteria* [2]. The gut microbiota plays an important role in maintaining homeostasis in the human gastrointestinal tract. In particular, the gut microbiota contributes to mucosal immunity and nutrient metabolism in the intestine. Firstly, as a host defense mechanism, the intestinal epithelium acts as a barrier between the luminal side and the lamina propria to prevent the entry of pathogens. The intestinal epithelial cells are connected by tight junctions to form a multifunctional cell complex [3]. The gut microbiota contributes to the restoration of the protein structure of the epithelial cell tight junctions [4]. In addition, the gut microbiota is involved in the production of defense-related molecules such as heat shock proteins [5,6] and mucin genes [7] in the intestinal epithelial cells. The gut microbiota also competes with pathogenic bacteria [8] and secrete antimicrobial peptides [9] to maintain homeostasis in the intestines. Secondly, the gut microbiota regulates the host’s intestinal mucosal immune system. Segmented filamentous bacteria, a type of intestinal bacteria, are associated with the differentiation and induction of interleukin-17-producing helper T cells, which are involved in inflammatory responses [10]. Clostridiales have also been shown to contribute to the differentiation and induction of regulatory T cells [11,12]. Thirdly, the gut microbiota is also related to nutrition. Enterobacteriaceae break down indigestible carbohydrates ingested by humans and produce short-chain fatty acids (SCFA) [13], which are used as an energy source by epithelial cells [14]. The gut microbiota also synthesizes vitamin K and B vitamins [15]. Thus, the gut microbiota plays many roles. In addition, an altered composition of the gut microbiota, termed dysbiosis, is linked to disrupted homeostasis and various diseases such as inflammatory bowel disease (IBD) [16,17], irritable bowel syndrome [18,19], asthma [20], and acute kidney injury (AKI) [21,22,23,24]. Multiple factors affect the composition of the gut microbiota. Concerning external factors, living area [25], eating habits [26], exercise [27], tobacco smoking [28], beverages, and drugs [29] are associated with the composition of the gut microbiota. Genetic background, age, sex, body mass index, and race are raised as internal factors that affect the gut microbiota [30,31,32]. In a study analyzing the gut microbiota in IBD, the results differed among countries [33], suggesting that comparison under varied circumstances may lead to inconsistent results. To understand the relationship between diseases and the gut microbiota, it would be preferable to compare within groups having the same background.

## 2. Pathologies Associated with Acute Kidney Injury and the Gut Microbiota

The pathogenesis of AKI is composed of various factors including inflammation, apoptosis, hemodynamic changes, and oxidative stress [34]. Although there are many reports on the relationship between kidney disease and the gut microbiota, the precise mechanisms are still unknown.

### 2.1. The Contribution of the Gut Microbiota

Germ-free (GF) mice are often used to explore the role of the gut microbiota in the pathogenesis of AKI. Kidneys of GF mice in steady state exhibited higher number of Natural Killer T cells and lower interleukin (IL)-4 levels compared with normal mice. GF mice with ischemia-reperfusion (I/R)-induced AKI showed exacerbated renal injury and increased CD8 T cells compared with those in normal mice. The transplantation of enterobacteria from normal mice into these GF mice resulted in reduced renal dysfunction and histological damage and a change in the composition of lymphocytes in the kidneys in the I/R model [35]. These findings suggest that the presence of the gut microbiota is involved in phenotypic changes of renal lymphocytes and alleviating renal damage in I/R, and that the gut microbiota have a renoprotective aspect. We also confirmed the renoprotective effects of the gut microbiota; I/R-induced renal injury was exacerbated in GF mice compared with that in normal mice, which was alleviated by stool transplantation from normal mice into GF mice [21]. Conversely, the depletion of the gut microbiota with antimicrobial agents reportedly protected against renal damage in the murine I/R model [36]. In mice with intestinal microbiota reduced by antimicrobial agent treatment, the number of F4/80 macrophages was decreased. In addition, the expression of the chemokine receptors C-X3-C motif chemokine receptor 1 (CX3CR1) and C-C motif chemokine receptor 2 (CCR2) on kidney F4/80+ macrophages and bone marrow monocytes was decreased in these gut bacteria-depleted mice. Furthermore, stool transplantation from normal mice worsened the kidney injury in the gut microbiota-depleted mice. Since the major findings were inconsistent between the reports, further studies are needed to elucidate the impact of gut microbiota on the pathogenesis of AKI.

### 2.2. SCFA and Inflammation

The gut microbiota produces SCFA, such as acetate, propionate, and butyrate, from indigestible dietary fibers. They are sources of energy for the body and are also involved in inflammation. It was reported that there are four receptors (GPR41, GPR43, Oflr78, and GPR109a) for SCFA in the kidney. GPR41 and GPR43 were identified in the kidney tissue and in the renal arteries [37], and the expression of GPR43 in the kidneys was increased by acetate administration in a mouse I/R model [38,39]. GPR43 was also expressed in granulocytes, monocytes, dendritic cells, and mast cells [40,41,42], suggesting that SCFA is involved in inflammation and immune responses. Oflr78 was expressed in the vascular smooth muscle and the juxtaglomerular apparatus, which regulates renin secretion in response to SCFA [37]. In Oflr78 knockout mice, antibiotic treatment reduced the biomass of the gut microbiota and an increase in blood pressure. This suggests that SFCA from the gut microbiota modulates blood pressure via Oflr78 [37]. Although GPR109a has been reported to be present in adipose tissue in the kidney, its function in kidney disease has not been clarified [43]. GPR109a stimulates the differentiation of regulatory T cells and interleukin 10-producing T cells [44], inhibits NF-κB activity, and induces apoptosis independent of histone deacetylases inhibition [45]. These are thought to be linked to the anti-inflammatory and tumor suppressive effects of butyrate. In the I/R-induced AKI model, it was reported that acetate directly acts on dendritic cells and tubular epithelial cells via GPR43, resulting in reduced kidney injury [38]. In a mouse model of lipopolysaccharide-induced AKI, acetate administration contributed to a decrease in T-cell histone deacetylase activity and reactive oxygen species, improved renal function, and reduced tubular damage [39]. In a murine model of Adriamycin-induced renal injury, butyrate from the gut microbiota was reported to mediate gene modification via GPR109a in renal epithelial cells, thereby reducing epithelial cell damage [46]. The nephroprotective effect of butyrate (reduced tubular damage) has also been reported in a contrast-induced AKI model in rats [47]. Because most of these findings were based on animal studies, the role of SCFA in the pathogenesis of human kidney diseases remains to be investigated.

### 2.3. Hemodynamics

Hypertension is often accompanied by AKI. It was reported that administration of angiotensin II did not cause an increase in blood pressure in GF mice, but increased blood pressure in normal mice with gut microbiota [48]. The reduced gene expressions of vascular monocyte chemoattractant protein 1 (MCP-1), inducible nitric oxide synthase (iNOS) and NADPH oxidase subunit Nox2 were demonstrated in GF mice treated with angiotensin II compared to normal mice, as well as a reduced upregulation of retinoic-acid receptor-related orphan receptor gamma t (Rorγt), the signature transcription factor for IL-17 synthesis [48]. These results led to an attenuated vascular leukocyte adhesion, and less infiltration of Ly6G(+) neutrophils and Ly6C(+) monocytes into the aortic vessel wall [48]. This suggested that the gut microbiota is involved in vascular immune cell infiltration and inflammation in angiotensin II administered mice, leading hypertension. In addition, it was reported that high-salt-feeding (2% NaCl in drinking water) mice developed renal damage and hypertension, and that fecal transplantation from these mice into normal mice resulted in leaky gut and hypertensive renal damage [49]. These results suggested that salt loading causes dysbiosis of the intestinal microbiota, which in turn triggers renal damage. Regarding this, the SCFA receptor was reportedly expressed in the juxtaglomerular apparatus, suggesting that gut microbiota-derived SCFA is involved in blood pressure regulation. Thus, the gut microbiota is likely to be associated with blood pressure.

### 2.4. Uremia and Dysbiosis

In patients with end-stage renal disease (ESRD), multiple factors such as uremia, diet, and drug use alter the composition of the gut microbiota. Therefore, the components of the intestinal microbiota differ between patients with ESRD and healthy subjects [50]. Interestingly, in patients with ESRD, DNA from gut bacteria was detected in the blood in addition to increased blood levels of D-lactate, high-sensitivity C-reactive protein, and IL-6 [51]. Moreover, it was reported that plasma endotoxin/lipopolysaccharide levels were increased in accordance with the chronic kidney disease (CKD) stage [52]. These data indicated that the gut microbiota and their products enter the circulation through the intestinal barrier and affect the immune system, leading to inflammation in CKD and ESRD. Although it is clear that AKI changes the components of the gut microbiota and its metabolites (Table 1), the precise mechanisms for the alteration of the gut microbiota in AKI need to be clarified.

### 2.5. Septic AKI

Sepsis is one of the most common risks for the development of AKI. 45 to 70% of AKI is reportedly associated with sepsis [53]. Septic AKI also alters composition of the gut microbiota, especially the loss of anaerobic bacteria. Systemic inflammation, antibiotics treatment, parenteral nutrition, and leaky gut may cause dysbiosis of the gut microbiota. In addition, decreased the gut microbiota and parenteral nutrition reduce the production of SCFA, which function as immune modulators [54].

## 3. Products Derived from the Gut Microbiota and AKI

### 3.1. D-Amino Acids

Amino acids have optical isomers, which can be distinguished into L- and D-forms, called chiral amino acids. In recent years, the technical innovation for measuring amino acids enabled us to separate and measure the chiral L- and D-forms with high accuracy [55]. The distributions and functions of D-amino acids were unclear until recently. In turn, chiral amino acid analysis revealed the existence of D-forms in vivo, and the kinetics and functions of D-amino acids in the body are now highlighted. In animal models, D-amino acid synthases and degrading enzymes are abundant in the brain and kidneys [56,57], suggesting that D-amino acids are metabolized in these organs. We have found that the stool of normal mice contains higher amounts of D-amino acids than that in the stool of GF mice [21], suggesting that D-amino acids are produced by the gut microbiota. In addition, we found that D-serine from intestinal bacteria showed reno-protective effects in a murine AKI model [21]. Nevertheless, a high concentration of D-serine induces renal damage [21,58,59], suggesting that the amount of D-serine is associated with specific responses, especially in the kidney. In addition to D-serine, various other kinds of D-amino acids are distributed in the organs. The pathophysiological roles of D-amino acids need to be elucidated.

### 3.2. Uremic Substances

Indoxyl sulfate (IS), p-cresyl sulfate (p-CS), and trimethylamine-N-oxide (TMAO) are known to be uremic substances associated with the gut microbiota. Studies with GF mice revealed that IS, p-CS, and TMAO are derived from the gut microbiota, and that diet is another source of TMAO [60]. The toxicity of IS and p-CS has been well studied in CKD, and their blood levels were associated with cardiovascular events and mortality in dialysis patients [61,62]. Similarly, elevated blood levels of TMAO were associated with the risk of cardiovascular disease and death [63]. In obese mouse models, TMAO-induced renal interstitial fibrosis was also reported [64]. It was reported that CKD patients with high blood TMAO levels had a poor life prognosis [65], and that dialysis patients with high TMAO serum levels had a higher risk of coronary events and death [66]. Additionally, recent systematic review and meta-analysis revealed a negative association between circulating TMAO concentrations and kidney function in CKD patients [67]. Uremic substances accumulated even in the AKI condition. The serum IS level was higher in AKI patients who required hospitalization than that in healthy subjects [68]. In this study, the mortality was increased in the group with higher serum IS levels [68]. Another report showed that serum IS and p-CS levels were elevated in patients with AKI and sepsis [69]. In both reports, it was also observed that serum levels of IS and p-CS were lower in AKI than in CKD [68,69]. Further studies are needed to elucidate the pathophysiological role of uremic substances in AKI.

## 4. Gut Microbiota as a Potential Therapeutic Option for AKI

Although some reports have raised gut microbiota and its products as therapeutic targets for CKD, the therapeutic potential still remains to be investigated in AKI patients. A recent systematic review shows that modifying the gut microbiota improves the prognosis of AKI [70]. Among the metabolites of the gut microbiota, SCFA and D-serine reportedly have renoprotective effects in AKI. These metabolites could be therapeutic options as postbiotics. In turn, some reports showed the therapeutic effects of probiotics. *Lactobacillus salivarius* BP121 attenuated cisplatin-induced tubular injury concomitant with reduced inflammation and oxidative stress [71]. A microbial cocktail of *Escherichia*, *Bacillus*, and *Enterobacter* reportedly protected against nephrotoxin-induced AKI [72]. ‘Prebiotics’ is a general term for nondigestible food ingredients (such as dietary fiber) that promote the growth of gut microbiota. Concerning prebiotics, their therapeutic effect in AKI has not been reported yet. However, a randomized clinical trial of probiotics and prebiotics in septic AKI was conducted [73], but the result is still not published. It was also reported that a high-fiber diet suppressed acute allograft rejection in a kidney transplant model [74].

Adsorbents are used to reduce uremic substances derived from the gut microbiota. AST-20 is effective in adsorbing precursors of uremic toxins such as indole and p-cresol. Although AST-20 reportedly reduced the decline of kidney function in CKD, the renoprotective effect in AKI is not clear. However, the lowering of uremic substances seems to have benefits even in AKI, based on the observational studies. Elevated levels of IS and p-CS in AKI were correlated with the severity of AKI (RIFLE classification) [75], and IS levels were associated with mortality in AKI, requiring hospitalization [68]. The suppression of IS and p-CS in the study of *Lactobacillus salivarius* BP121 mentioned previously suggested that the suppression or adsorption of these uremic substances in AKI may be a therapeutic target. In addition, rifaximin, a kind of nonabsorbable antibiotic, decreased the TMAO blood level in a mouse model of CKD [76]. Moreover, rifaximin treatment reduced the incidence of AKI, hepatorenal syndrome, and the need for renal replacement therapy in a human retrospective study [77]. The pathophysiologic roles of uremic substances need to be investigated in the AKI condition.

## 5. Conclusions

The relationship between AKI and the gut microbiota is still poorly understood (Figure 1), unlike the relationship between CKD and the gut microbiota. Further studies on AKI pathogenesis and therapeutic applications are expected.

## Figures and Tables

**Figure 1 toxins-13-00369-f001:**
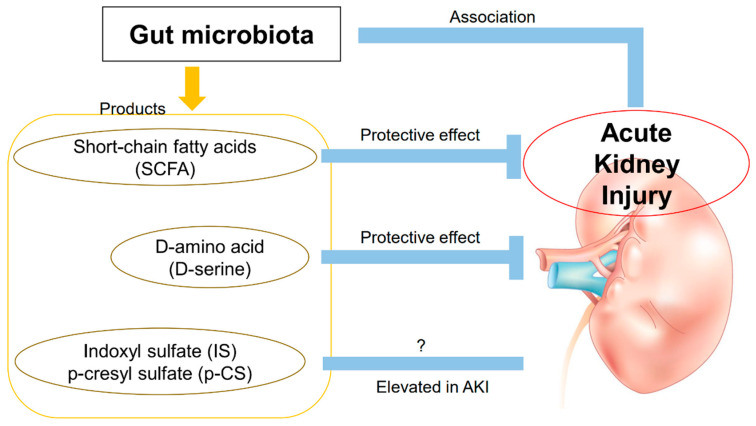
A brief summary of AKI and the gut microbiota. The gut microbiota is involved in the pathogenesis of AKI. SCFA and D-serine, products of the gut microbiota, have renoprotective effects against AKI. IS and p-CS are elevated in AKI, but their effects on AKI are unknown.

**Table 1 toxins-13-00369-t001:** AKI model and gut microbiota analysis.

Author	Type of Animal	AKI Method	Bacterial Collection Method, Period	Bacterial Storage Method	Bacterial Analysis Methods	Changed Bacteria	Changed Metabolites
Nakade [21]	C57BL/6 mice	I/R Unilateral 40 min	Feces Day 0, 2, 10 after I/R	−80 °C freezer	16S rRNA gene-sequencing analysis	Increased *Lactobacillus,* *Clostridium,* *Ruminococcus* Decreased *Bifidobacterium* TM7	Increased D-Serine/L-Serine
Yang [22]	C57BL/6 mice	I/R Bilateral 25.5 min (SPF mice) 28.5 min (GF mice)	Feces Day 1, 3, 7 after I/R	−70 °C freezer	16S rRNA gene-sequencing analysis	Increased *Enterobacteriacea* Decreased *Lactobacilli* *Ruminococacceae*	Decreased SCFA
Andrianova [23]	Wistar rats	I/R Unilateral 40 min	Faces Period is not mentioned	Not mentioned	Metagenomic analysis	Increased *Staoh vlococc* *Prevotella*	Increased 32 acylcarnitines Decreased tyrosine, tryptophan, proline
Samanta [24]	Wistar rats	Hypoxia environment Barometric pressure: 7.3, 9.3, 11.8 psia	Feces Day0, 7 after hypoxia	Cultured	Limited bacterial analysis	Increased *Escherichia coli* *Bacteroidetes* *Bifidobacterium* *Salmonella*	Not mentioned

Abbreviations: AKI, acute kidney injury; GF, germ free; I/R, ischemia reperfusion; SPF, specific pathogen free.

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
