# Peer review of "Significance of the Gut Microbiota in Acute Kidney Injury"

_toxins, 2021, doi:10.3390/toxins13060369_

Round 1
Reviewer 1 Report
The current manuscript entitled “Significance of the Gut Microbiota in Acute Kidney Injury “ depics the recent studies on the role of the gut microbiota in the pathogenesis of acute kidney injury. Soluble mediators of microbiome effects (SCFA, aminoacids, uremic solutes) are depicted in relation to AKI, however, intestinal barrier and immunologic are not given the credit they deserve, as contributing factors in AKI. In the past year there are been a few numbers of reviews (Gong, J, Seminars in nephrology vol. 39,1 (2019): 107-116, PMID: 30606403 and Rabb H, Pluznick J, Noel S: The Microbiome and Acute Kidney Injury. Nephron 2018;140:120-123. doi: 10.1159/000490392 and Gharaie S, Noel S, Rabb H: Gut Microbiome and AKI: Roles of the Immune System and Short-Chain Fatty Acids. Nephron 2020;144:662-664. doi: 10.1159/000508984) covering the microbiota-kidney axis in AKI, and this review follows on their footsteps. Besides the addition of 5 recent publications ( 4 of 2020, and 1 of 2019), in many instances the authors to not discuss in depth the implications of the newly published studies and the relation to previous works and resolve by stating that “the precise mechanisms for the alteration of the gut microbiota in AKI need to be clarified”, “the pathophysiological roles of D-amino acids need to be elucidated”, “Further studies are needed to elucidate the pathophysiological role of uremic substances in AKI.” and “The pathophysiologic roles of uremic substances need to be investigated in the AKI condition“. To conclude, I have the impression that the review has been written in a rush, following the line of thought of previous reviews, and by the lack of discussion of recent studies, it does not covers novelty.
The current manuscript entitled “Significance of the Gut Microbiota in Acute Kidney Injury “ depicts the recent studies on the role of the gut microbiota in the pathogenesis of acute kidney injury. Soluble mediators of microbiome effects (SCFA, aminoacids, uremic solutes) are depicted in relation to AKI, however, intestinal barrier and immunologic are not given the credit they deserve, as contributing factors in AKI. In the past year there are been a few numbers of reviews (Gong, J, Seminars in nephrology vol. 39,1 (2019): 107-116, PMID: 30606403 and Rabb H, Pluznick J, Noel S: The Microbiome and Acute Kidney Injury. Nephron 2018;140:120-123. doi: 10.1159/000490392 and Gharaie S, Noel S, Rabb H: Gut Microbiome and AKI: Roles of the Immune System and Short-Chain Fatty Acids. Nephron 2020;144:662-664. doi: 10.1159/000508984) covering the microbiota-kidney axis in AKI, and this review follows on their footsteps. Besides the addition of 5 recent publications ( 4 of 2020, and 1 of 2019), in many instances the authors to not discuss in depth the implications of the newly published studies and the relation to previous works and resolve by stating that “the precise mechanisms for the alteration of the gut microbiota in AKI need to be clarified”, “the pathophysiological roles of D-amino acids need to be elucidated”, “Further studies are needed to elucidate the pathophysiological role of uremic substances in AKI.” and “The pathophysiologic roles of uremic substances need to be investigated in the AKI condition“. To conclude, I have the impression that the review has been written in a rush, following the line of thought of previous reviews, and by the lack of discussion of recent studies, it does not covers novelty.
Please see below some suggestions to address:
Abstract:
I would include more about microbiome as this is the main topic of the paper, what is currently in the abstract is not giving much information about what will be in the article
Overview of the Gut Microbiota
Introduction: it is not flowing nicely. Sentences are not really connected to each other and it makes it a bit difficult to follow.
Page 1, line 22-23 “They also compete with pathogenic bacteria [8] and secrete antimicrobial peptides [9] to maintain the intestinal environment.”
What is “they”? and what does “to maintain the intestinal environment” mean exactly? I think this is not very precise and it is confusing
Page 1, line 32-34“An altered composition of the gut microbiota, termed dysbiosis, is linked to disrupted homeostasis and various diseases such as inflammatory bowel disease (IBD) [16,17], irritable bowel syndrome [18,19], and asthma [20].”
I would add also AKI here since the paper is about AKI
Page 1, line 35-37 “Multiple factors affect the composition of the gut microbiota. As for external factors, country and region [21], eating habits [22], exercise [23], tobacco smoking [24], beverages, and drugs [25] are associated with the composition of the gut microbiota”
What is meant exactly with “country and region”?
Pathologies Associated with Acute Kidney Injury and the Gut Microbiota
Page 2, line 45 “Acute kidney injury (AKI) involves a complex interplay of pathophysiology, including inflammation, apoptosis, hemodynamic changes, and oxidative stress [30].”
I don’t think “complex interplay of physiology” means something. I would rephrase this.
The Contribution of the Gut Microbiota
Page 2, line 55-56 “These results suggested that the gut microbiota may play a role in renal protection.”
I think this sentence should be expanded further. It Is more complicated than just “the microbiome plays a role in renal protection”
Page 2, line 56-59 “We also confirmed the renoprotective effects of the gut microbiota; I/R-induced renal injury was exacerbated in GF mice compared with that in normal mice, which was alleviated by stool transplantation from normal mice into GF mice [32]. Conversely, the depletion of the gut microbiota with antimicrobial agents reportedly protected against renal damage in the murine I/R model [33] in mice with intestinal microbiota reduced by antimicrobial agent treatment, the number of F4/80 macrophages was decreased. In addition, the expression of the chemokine receptors C-X3-C motif chemokine receptor 1 (CX3CR1) and C-C motif chemokine receptor 2 (CCR2) on kidney F4/80+ macrophages and bone marrow monocytes was decreased in these gut bacteria-depleted mice. Furthermore, stool transplantation from normal mice worsened the kidney injury in the gut microbiota-depleted mice. This result suggested that the presence of the gut microbiota has a negative effect on renal damage. Although the major findings were inconsistent between the reports, they suggested that the gut microbiota is related to the pathogenesis of AKI.”
The laying down of these two studies does not reflect the conclusion.
SCFA and Inflammation
Page 2, line 71“The intestinal microflora produces SCFA, such as acetate, propionate, and butyrate, from indigestible dietary fibers.”
I wouldn’t use the word microflora. Be consistent.
Page 2, line 80 “In Oflr78 knockout mice, antibiotic treatment caused dysbiosis of the gut microbiota and an increase in blood pressure [34].”
Can you provide an explanation for that?
Hemodynamics
Page 3, line 103 “In addition, it was reported that salt-loaded mice developed renal damage and hypertension, and that fecal transplantation from these mice into normal mice resulted in leaky gut and hypertensive renal damage [46].”.
An explanation of the “salt loaded” model would benefit the reader salt loaded mice?
Page 3, line 101 “This suggested that the gut microbiota is involved in the enhancement of angiotensin II-induced blood pressure elevation”
Can you further discuss this?
Page 3, line 107 “Regarding this, the SCFA receptor was reportedly expressed in the juxtaglomerular apparatus, suggesting that gut microbiota-derived SCFA has some roles in the cell.” This sentence is not very scientific. What does “some role’ means? Further elaborate.
Gut Microbiota as a Potential Therapeutic Option for AKI
Page 5, line 173 “However, a randomized clinical trial of probiotics and prebiotics in septic AKI was conducted.”
It would be advised to explain at least what they found in the study.
Reviewer 2 Report
The topic of the paper is very interesting and little studied.
The arguments that are exposed are clear but do not connote a real clinical application. Given theAKI importance in critically ill patients, it would be appropriate to proceed with a clinical and practical connotation of the review.
In the title they speak of AKI, while in the text of both AKI and CKD are analyzed in term of gut microbiota, it would be appropriate to evaluate the title by introducing a more generic one that includes both: "Significance of the Gut Microbiota in Kidney Injury" or " Gut Microbiota in Kidney Injury".
Another interesting part of the microbiota during AKI should be considered: septic AKI and the role of antibiotic therapies on the microbiota, and their interactions.
Reviewer 3 Report
In this review article, the author concisely and comprehensively illustrated the significant role of microbiota on acute kidney injury (AKI) pathogenesis.
This review covered the overview of the gut microbiota and the major topics of recent progress gut microbiome analysis in human AKI and animal models. The author reviewed the short chain fatty acid(SCFA) hemodynamics(especially salt and hypertension), and uremic solutes(indoxyl sulfate(IS), p-cresyl sulfate(PCS) and trimethylamine-N-Oxide(TMAO)).
Therapeutic potentials of gut microbiota as the target of modification of dysbiosis in AKI were mentioned.
Reviewer 4 Report
The title of the article is “Significance of the Gut Microbiota in Acute Kidney Injury”.
The authors conducted a narrative review. This review aimed to updated evidence of the role of the gut microbiota in the pathogenesis of AKI.
This is an interesting review that can benefit from more thorough review and appears to be well performed in general and the manuscript is well written. This review may help clinician-researcher and scientist to a better understanding of the relationships between the intestinal microbiome and renal physiology/pathophysiology and applying this knowledge for design the future therapeutic trials. However, the manuscript still could be further improved after some revisions.
Specific comments:
- The outline of this review had similar to “PMID: 30606403”. A several updated evidence from Ref.32, 50, 51, and 52 were found from this review. Please add more updated evidence.
Reference
- Gong J, Noel S, Pluznick JL, Hamad ARA, Rabb H. Gut Microbiota-Kidney Cross-Talk in Acute Kidney Injury. Semin Nephrol. 2019 Jan;39(1):107-116. doi: 10.1016/j.semnephrol.2018.10.009. PMID: 30606403; PMCID: PMC6322425.
- Nowadays, in most evidence hierarchies current, well designed systematic reviews and meta-analyses are at the top of the pyramid. Please provide findings from a recent systematic review and meta-analysis. Please update the potentially relevant publications.
Reference
- Rydzewska-Rosołowska A, Sroka N, Kakareko K, Rosołowski M, Zbroch E, Hryszko T. The Links between Microbiome and Uremic Toxins in Acute Kidney Injury: Beyond Gut Feeling-A Systematic Review. Toxins (Basel). 2020 Dec 11;12(12):788. doi: 10.3390/toxins12120788. PMID: 33322362; PMCID: PMC7764335.
Additionally, please including more recently RCT in your review.
If the above suggestions are incorporated and the paper is thoroughly edited, it will be a strong contribution to the literature. Thank you so much to let me have opportunity to work with this research article.
Round 2
Reviewer 1 Report
Accept in current form
Reviewer 2 Report
no suggestions
Reviewer 4 Report
The authors addressed all my previous concerns. I have no additional comment.